# The Role of Empathic Communication in the Relationship between Servant Leadership and Workplace Loneliness: A Serial Mediation Model

**DOI:** 10.3390/bs14010004

**Published:** 2023-12-20

**Authors:** Jiaying Jin, Hiroshi Ikeda

**Affiliations:** 1Graduate School of Human-Environment Studies, Kyushu University, Fukuoka 8190382, Japan; 2Faculty of Human-Environment Studies, Kyushu University, Fukuoka 8190382, Japan

**Keywords:** workplace loneliness, servant leadership, empathic communication, behavioral empathy, employee well-being

## Abstract

Researchers have increasingly concentrated on loneliness in the workplace as a crucial factor influencing the mental health of employees and the viability of telework. In contrast, the current understanding of the strategies mitigating workplace loneliness and how leaders utilize their behaviors to impact followers’ loneliness remains limited. Since servant leadership values the emotional needs of followers and displays a high level of empathy, this study investigated the direct and indirect effects of servant leadership on workplace loneliness. In this study, 267 employees (mean age = 31.5 years) from 28 provinces in China were recruited to participate in this survey. We proposed that servant leaders motivate their own empathic communication and other followers’ empathic communication to reduce lonely followers’ workplace loneliness. This research further examined the relationship between the leader’s and colleagues’ empathic communication, and the two jointly mediate the connection between servant leadership and followers’ workplace loneliness. We constructed a serial mediation model to examine the relationships between servant leadership, leader’s empathic communication, colleagues’ empathic communication, and workplace loneliness. The results indicate that servant leadership creates a cycle of empathy and provides insights into building a culture of empathy to improve employee well-being.

## 1. Introduction

Loneliness is dissatisfaction with the inconsistency between ideal and existing relationships [1]. Workplace loneliness is an extension of loneliness in organizational research. During COVID-19, the popularity of telework increased significantly, and workplace loneliness has received considerable attention as a significant factor influencing the job engagement of teleworkers [2,3]. Workplace loneliness reduces followers’ performance, hurts their colleagues (unfriendliness) and organization (decreased emotional commitment), and causes low quality of leader–member exchange and less organizational citizenship behavior [4,5]. Wright (2005) [6] has supposed that workplace loneliness occurs when there is a discrepancy between the quantity and quality of relationships people expect and have at work, and they are incapable of compensating for this discrepancy. Employees may feel lonely regardless of organizational hierarchy [7]. Fostering satisfying relationships within an organization to improve followers’ performance and well-being poses a serious management challenge.

However, developing relationships at work can be challenging. Unfavorable workplace conditions, including work overload, job burnout, and workplace ostracism, can increase followers’ workplace loneliness [8,9,10]. Overwork exacerbates followers’ short-term stress and deteriorates their social relationships [11]. An International Labor Organization (ILO) report found that over one-third of all workers worldwide regularly exceeded 48 h per week in 2019 [12]. Moreover, Since the COVID-19 pandemic, most employees work at various times and locations, making workplace loneliness a problem due to social isolation and decreased face-to-face communication. There is little quantitative research to date on reducing followers’ workplace loneliness at the organizational level. Elche et al. [13] call for further research on how organizations help followers alleviate workplace loneliness, and this study aims to answer this call.

In contrast to general loneliness, workplace loneliness is easily impacted by the relationship between the organization’s members, especially between the leaders and the followers [10]. Extensive evidence also suggests that leadership can influence interpersonal relationships at work [14,15,16]. Servant leadership was introduced by Greenleaf (1977) [17], which emphasizes service to others and puts the interests of followers above leaders’ interests. This study examined the connection between servant leadership, behavioral empathy (empathic communication), and followers’ workplace loneliness. We provided new insights into how to decrease followers’ workplace loneliness and demonstrated the effectiveness of servant leadership in maintaining followers’ mental health.

## 2. Literature Review and Hypotheses

### 2.1. Workplace Loneliness

Loneliness is a detrimental and shared human experience [18]. Nevertheless, it has not been properly tackled in the context of organizations. Based on the belongingness hypothesis, individuals have an innate desire to develop and maintain a certain number of long-lasting, positive, and valuable relationships [19]. Failure to meet the need to belong and dissatisfaction with interpersonal relationships may lead to feelings of loneliness [20]. Much literature considered social isolation and loneliness essential to physical and mental health [21,22]. However, social isolation and loneliness are fundamentally different. Social isolation primarily refers to the absence of social connections between individuals, which can be assessed using objective indicators, such as living alone [1,23,24]. On the contrary, loneliness is dissatisfaction with the disparity between ideal and actual social relationships, accentuating the individual’s subjective feelings [25,26]. A person with an extensive social network may still experience loneliness.

As a subjective experience, the antecedents of loneliness vary by context, environment, and condition [27]. Therefore, researchers have more precisely defined loneliness in the workplace. Wright et al. (2006) [28] proposed that workplace loneliness is the suffering induced by the perception of a shortage of high-quality interpersonal relationships between coworkers. Ozcelik and Barsade (2018) [4] suggested that workplace loneliness is employees’ subjective feelings and thoughts about whether their coworkers and leaders fulfill their affiliation needs. Thus, most of the time, negative interactions with leaders or coworkers are to blame for feelings of loneliness suffered at work.

Regarding the dimensions of loneliness, Russell (1980) [29] considered loneliness as a single structure, and other researchers investigated loneliness in multiple dimensions [30]. Wright et al. (2006) [28] divided the items into two subcategories describing the social companionship and emotional deprivation of workplace loneliness. In particular, inadequate social companionship relates to an employee’s disengagement from the organization’s network of relationships and generates a sense of alienation from other organization members. Emotional deprivation occurs when employees’ need to belong is not met due to quantitative or qualitative shortcomings in their interpersonal relationships within the organization. In terms of measurement, the two-factor structure of workplace loneliness is generally recognized [31].

Even though workplace loneliness is a common experience for employees [32], we know little about how to combat loneliness in the workplace effectively. The lack of friends is viewed as a social failure [33]. Consequently, loneliness is frequently stigmatized and trivialized [18,26]. People may attempt self-masking to conceal their lonely experience [34] and resort to social avoidance rather than constructing social relationships in the workplace [5]. Similarly, refusing to self-disclosure hinders the normal development of interpersonal relationships among lonely individuals [35]. Due to the difficulty of fleeing this vicious cycle alone, it makes sense to investigate leadership to mitigate followers’ workplace loneliness.

### 2.2. Servant Leadership and Workplace Loneliness

Although Greenleaf [17] did not provide a precise definition of servant leadership [36], there are common characteristics of servant leadership in subsequent studies. The primary motivation behind servant leadership is the desire to serve [17,37,38]. Servant leadership prioritizes the needs of its followers over those of the organization and themselves [39], emphasizing followers’ personal growth [40,41]. In contrast to leadership that emphasizes improving organizational goals and performance, servant leadership focuses on interaction with followers and considers their well-being, emotions, and ethics [42,43]. Servant leadership enables followers to be more engaged and productive by placing a premium on followers’ mental health and personal growth [44]. Servant leadership could establish trust relationships with followers [45] and improve leader–member exchange (LMX) quality [46], which makes followers under servant leadership experience higher job satisfaction, employee well-being, and less burnout [47,48,49].

Since the appropriate amount of relationships varies from person to person, the quality of relationships may be more significant than quantity in preventing and alleviating workplace loneliness [5]. Despite growing evidence that leadership is critical to curbing workplace loneliness, how servant leadership affects followers’ workplace loneliness is still being determined. In terms of followers’ social well-being, servant leadership values long-term relationships with followers and is sensitive to followers’ emotional needs [50,51], which contributes to a positive team atmosphere among followers [52]. For instance, servant leadership can satisfy followers’ need to belong by fostering an inclusive climate within the organization and promoting open and honest communication to allow followers to express their authentic selves [43,53]. Given the above arguments, servant leadership can potentially enhance the quality of relationships experienced by lonely workers. Therefore, followers who follow servant leadership may experience less workplace loneliness.

**Hypothesis** **1.**
*Servant leadership is negatively related to followers’ workplace loneliness.*


### 2.3. Empathy

Empathy is an essential characteristic of servant leadership [54], and higher empathy may explain servant leaders’ sensitivity to followers’ emotional needs. Most researchers have considered empathy a multidimensional concept, with affective and cognitive empathy being the most common distinctions [55]. Affective empathy is defined as emotional congruence between the target person and observer [56,57,58], and cognitive empathy refers to the comprehension of others’ mental and emotional states [59,60]. Cognitive empathy is necessary for socializing to perceive others’ internal states [61], representing an imaginative understanding of others’ situations [62,63].

Besides comprehension of the target’s internal state, empathy is also reflected in behavior. Empathy implies understanding others and expressing understanding to others [55,64]. Multiple processes are involved in empathy, including identifying and comprehending others’ mental states and responding with appropriate behaviors [65]. The research review has suggested three types of empathy: affective, cognitive, and behavioral [59]. Behavioral empathy is the external manifestation of empathy, and in contrast to cognitive and affective empathy, focuses on the target person [66]. Behavioral empathy is the ultimate consequence of the entire empathy process.

Different communication styles influence the target’s perception of empathy [67]. As a form of behavioral empathy, Empathic communication indicates that the observer intentionally expresses an understanding of the target person’s internal state based on emotional and/or cognitive empathy, including inquiries, rhetorical questions, and nonverbal behaviors (e.g., eye contact, concerned facial expressions) [59,68,69,70]. Through empathic communication, individuals can consciously help others [71]. Extensive research has been conducted on the effectiveness of empathic communication in doctor-patient interactions [70,72,73]. However, the impact of empathic communication on fostering workplace relationships still needs to be verified. Applying empathic communication may be an improved method for combating loneliness.

#### 2.3.1. Leader Empathy

Followers perceive leadership via leaders’ emotional and mental abilities, and empathy can promote followers’ perceptions of leadership [74]. Spears [75] summarized ten traits related to servant leadership: listening, empathy, healing, awareness, persuasion, conceptualization, foresight, stewardship, commitment to the development of followers, and community building. Empathy is a valuable leadership characteristic [76], yet it is considerably undervalued compared to other leadership traits, such as responsibility and passion [77]. Empathy reveals leaders’ authentic concern for their followers’ needs and interests [78], and followers also desire an emotional connection with their leaders through empathy [79].

As an interpersonal helping strategy, empathy fosters positive leader–follower relationships and high-quality LMX [80,81]. Followers perceive servant leadership as possessing strong empathy [82], which may explain servant leaders’ sensitivity to followers’ needs [50]. Empathy is the foundation for leaders’ relations-oriented behaviors and enables leaders to choose the most effective behaviors to meet followers’ requirements [81]. Servant leaders are more capable of alleviating followers’ pain by incorporating empathy as a core leadership skill [83]. The qualitative research confirmed that servant leaders’ empathy with their followers’ suffering motivates their perspective-taking and compassionate response to help followers maintain emotional balance [84]. Accordingly, servant leaders may utilize empathic communication to alleviate followers’ workplace loneliness. The hypotheses are as follows:

**Hypothesis** **2.**
*Leaders’ empathic communication mediates the relationship between servant leadership and followers’ workplace loneliness.*


#### 2.3.2. The Mediating Effect of Colleagues’ Empathy

Servant leaders trust that serving their followers improves the organization’s long-term success [85]. Assisting followers in their personal growth and development is a crucial characteristic of servant leadership [50,86], as it encourages followers to act in ways that benefit the organization [87]. Servant leadership is dedicated to cultivating morality in their followers, motivating followers to serve, and advancing the common good, and their ultimate objective is for followers to become servants [88].

Servant leadership develops a cycle of service [49]. Servant leaders inspire their followers to perform actions to support colleagues in distress at both individual and organizational levels. Based on social learning theory [89], because followers regard servant leaders as trusted role models [45,90], they aspire to imitate leaders’ behaviors [91] and desire to support colleagues in distress [92]. Following social exchange theory [93] and norm or reciprocity [94], servant leadership shows concern about followers’ well-being, which makes followers more devoted to their leader and organization and obligingly aids their colleagues to compensate for these positive leader behaviors [95]. At the organizational level, servant leadership creates an environment conducive to collaborative support by emphasizing group identity and facilitating social exchange relationships between followers [51,95]. Specifically, servant leadership fosters a variety of positive organizational climates (e.g., inclusive climates and service climates), indirectly encouraging followers’ organizational citizenship behavior [43,86,96].

Shaping followers’ empathy is vital for motivating followers’ service behaviors and organizational citizenship behavior [13]. Followers with strong empathy are more likely to help their lonely colleagues [97]. Empathy training shows that empathic expression is a teachable communication skill [98,99,100], and cognitive empathy significantly increased after training [101].

Overall, workplace loneliness can have negative consequences for organizations and individuals. The core of servant leadership is serving employees, which helps servant leaders meet the emotional needs of lonely followers and directly mitigates followers’ workplace loneliness. Behavioral empathy is an outgrowth of the servant leaders’ servant spirit. Empathic communication, as a form of behavioral empathy, mediates the relationship between servant leadership and followers’ workplace loneliness and indirectly reduces loneliness.

Empathic communication happens not only from servant leaders but also from lonely followers’ colleagues. Servant leaders cultivate colleagues’ empathic communication in three ways. Firstly, based on social exchange theory [93], followers working under servant leaders develop a quality exchange relationship with servant leaders by offering to help lonely colleagues in return for servant leaders. Additionally, social learning theory [89] indicates that followers perceive servant leaders as role models and learn empathic communication with their colleagues from servant leaders. Moreover, servant leadership fosters a service climate within the organization, encouraging more followers to serve other organization members.

In the present study, social exchange theory [93], social learning theory [89], and the cycle of service created by servant leadership [43,96] collectively supported the serial mediation model. Leader’s empathic communication and colleagues’ empathic communication individually or conjointly mediated the negative correlation between servant leadership and workplace loneliness. Therefore, we predict the following: 

**Hypothesis** **3.**
*Colleagues’ empathic communication mediates the relationship between servant leadership and followers’ workplace loneliness.*


**Hypothesis** **4.**
*Servant leadership indirectly reduces followers’ workplace loneliness via the chain of leaders’ empathic communication and colleagues’ empathic communication.*


#### 2.3.3. Self-Report and Other-Report

Empathy assessments can be classified as the observer’s empathic experiences (self-report) and perceived empathy by their partner (other-report). There has been little agreement on the assessment of empathy, given that empathy scores differ based on the assessor [102,103]. Despite the widespread use of self-reports, recent research has demonstrated that self-reported cognitive empathy only explains 1% of the variance in empathic behavior in interpersonal interactions [104].

The self-reported deficits in the explanatory power of empathic behavior might be driven by inadequate empathic accuracy [105] and bias in self-perception [106]. Most research on empathic accuracy focuses on the observer (the person making the judgment) and ignores the impact of the target person on empathy accuracy [107,108]. Most researchers have used self-report because cognitive empathy and affective empathy are intrinsic psychological processes. However, behavioral empathy prioritizes the target person’s true feelings about empathic behavior, making other-report more desirable. In addition, individuals are incapable of making accurate assessments [109,110] and may exaggerate their abilities [106]. Accordingly, this study used other-report questionnaires (reported by followers) to measure empathy from supervisors and colleagues. We developed a serial mediation model (Figure 1).

## 3. Materials and Methods

We applied the back translation method and made the pilot test to verify that the scales were appropriate in the Chinese context. The participants of the actual survey were 267 employees from different companies in China. All measured variables were measured on 5-point Likert scales, ranging from 1 (strongly disagree) to 5 (strongly agree). In this study, Likert-scale data were handled as scale data.

Concerning the survey design, we asked participants to recall their immediate leader while reporting on the servant leadership and the leader’s empathy communication. Participants were asked to respond to the leader with whom they were most familiar if they had more than one immediate leader. These instructions clarified and fixed the participant’s evaluation object (the leader). To ensure the generalizability of the findings, we recruited employees from 28 provinces in China to participate in the survey, and the sample was gender-balanced (male 59.6%, female 40.4%).

For servant leadership, employees’ perceptions of servant leadership were measured using the 7-item servant (SL-7) leadership scale [111]. An example item is as follows: “I would seek help from my leader if I had a personal problem”. The scale had satisfactory fit values for a single-factor structure (χ^2^(14) = 33.604, *p* < 0.05; χ^2^/df = 2.4; RMSEA = 0.073; RMR = 0.048; CFI = 0.965; NFI = 0.942) and Cronbach’s α = 0.82.

For leader’s empathic communication, employees reported their leader’s empathic communication by using 6 items developed by Nicolai et al. (2007) [112]. This scale was developed to measure empathic communication in physician-patient interactions. We deleted 3 items that were not related to the workplace environment and changed “physician” to “my leader” in other items. Participants reported their leader’s empathic communication by 6 items. A given item was: “My leader treats me as an equal partner in communication”. The CFA results showed support for the integrity of the scale for a unidimensional solution (χ^2^(9) = 15.864, *p* > 0.05; χ^2^/df = 1.763; RMSEA = 0.054; RMR = 0.021; CFI = 0.992; NFI = 0.982) and Cronbach’s α = 0.9.

For colleagues’ empathic communication, we also applied the 6 items developed by Nicolai et al. (2007) [112] to test participants’ perceptions of communication with their colleagues. The CFA showed satisfactory fit (χ^2^(8) = 22.719, *p* < 0.05; χ^2^/df = 2.84; RMSEA = 0.083; RMR = 0.02; CFI = 0.969; NFI = 0.954). Cronbach’s α for this scale was 0.81.

For workplace loneliness, followers’ work loneliness was measured by Wright et al. (2006) [28] with 16 items. One item was omitted since there was a low loading (<0.4) on the social companionship factor. A CFA for a two-factor model showed satisfactory fit (χ^2^(77) = 157.34, *p* < 0.05; χ^2^/df = 2.043; RMSEA = 0.063; RMR = 0.025; CFI = 0.946; NFI = 0.902). Cronbach’s α for emotional deprivation and social companionship subscales were 0.85 and 0.81.

## 4. Results

Before performing data analysis, we utilized Harman’s single factor score to test for common method bias. The result of exploratory factor analysis showed that the total variance for the first principal component is 36.56% (less than 40%), suggesting no serious problem with CMB. SPSS28.0 was used to test the hypothesis.

### 4.1. Descriptive Statistics and Correlation Analysis

Table 1 shows the demographic characteristics of the participants: 159 (59.6%) were males, and 108 (40.4%) were females. Participants’ mean age was 31.5 years (SD = 5.05). We questioned participants about their tenure, positions, and other variables to assess the test’s generalizability.

Correlation coefficients for each variable are shown in Table 2. Servant leadership was negatively correlated with followers’ workplace loneliness (r = −0.50, *p* < 0.01). There was a significant positive correlation between servant leadership and leader’s empathic communication (r = 0.80, *p* < 0.01) and a significant negative correlation between servant leadership and workplace loneliness (r = −0.64, *p* < 0.01). Moreover, leader’s empathic communication was positively correlated with colleagues’ empathic communication (r = 0.56, *p* < 0.01). The results of the correlation analysis provide preliminary support for the serial mediation model. Leader’s empathic communication and colleagues’ empathic communication may be two variables that mediate the negative correlation between servant leadership and workplace loneliness.

### 4.2. Hypothesis Testing

We performed hierarchical regression with age, tenure, and position of participants as control variables to assure the accuracy of the results. In Table 3, servant leadership positively predicted leaders’ empathic communication (β = 0.85, *p* < 0.001) and colleagues’ empathic communication (β = 0.28, *p* < 0.001), and leaders’ empathic communication positively predicted colleagues’ empathic communication (β = 0.13, *p* < 0.05). Since the direct effect of servant leadership on workplace loneliness was not significant, servant leadership can not directly alleviate followers’ workplace loneliness. Hypothesis 1 was not supported.

To test Hypotheses 2 to 5, we utilized a serial mediation model and the bootstrap approach to investigate the mediation effect of leaders’ and colleagues’ empathic communication between servant leadership and workplace loneliness. Table 4 shows that all three mediation pathways were significant, with a total mediating effect of −0.35 (SE = 0.06, 95% CI = [−0.470, −0.249]). The mediating effect of path 1 was −0.19, accounting for 52.40% of the total indirect effect (SE = 0.06, 95% CI = [−0.304, −0.082]), and the mediating effect of path 2 is −0.12, accounting for 33.59% of the total indirect effect (SE = 0.03, 95% CI = [−0.183, −0.061]). There is no significant difference between path 1 and path 2 (SE = 0.07, 95% CI = [−0.334, 0.110]). By comparing the mediating effect of empathic communication from leaders and colleagues, we identified that the mitigating effect on followers’ workplace loneliness was the same regardless of who initiated the empathic communication. This result confirms the general effectiveness of empathic communication in the workplace.

The serial mediation effect of leaders’ empathic communication and colleagues’ empathic communication on followers’ workplace loneliness was −0.05 (SE = 0.03, 95% CI = [−0.334, −0.003]). Hypotheses 2–4 were supported. Figure 2 depicts the final serial mediation model.

## 5. Discussion

This study aimed to investigate the positive effects of servant leadership on workplace loneliness. We introduced empathic communication as a strategy for servant leadership to reduce workplace loneliness and tested the mediation effect of leaders’ and colleagues’ empathic communication on the relationship between servant leadership and followers’ workplace loneliness. According to social learning theory [89] and social exchange theory [93], servant leaders can apply empathic communication with lonely followers and serve as role models to facilitate empathic communication among all followers. Servant leadership influences the followers’ workplace loneliness via three paths: (1) leader’s empathic communication, (2) colleagues’ empathic communication, and (3) the serial mediation of leader’s empathic communication and colleagues’ empathic communication. This study proved the effectiveness of servant leadership in maintaining followers’ mental health and explored the internal mechanisms by which servant leadership affects followers’ workplace loneliness.

### 5.1. The Mediation Role of Leaders’ Empathic Communication

Although prior research indicated that leadership decreases followers’ negative emotions [113], in contrast to transformational leadership, servant leadership cannot directly influence workplace loneliness [114]. Servant leadership values the followers’ emotional needs and carries empathic traits [54], and they may utilize empathy to satisfy lonely followers’ need to belong [115]. Empathic communication refers to leaders intentionally expressing understanding to build emotional ties with their followers [116]. Through empathic communication, servant leadership improves followers’ perceptions of interpersonal relations in the workplace. The results suggested that leaders’ empathic communication mediates the relationship between servant leadership and followers’ workplace loneliness, and servant leadership can indirectly ease followers’ loneliness experience. This finding further supports the effectiveness of empathy in leadership and the fact that a leader’s empathy positively predicts followers’ well-being [80].

Empathic communication emphasizes the leader’s empathic behavior rather than internal traits. The mediation effect of leaders’ empathic communication provides additional evidence that the primary factor influencing followers is the leaders’ empathic behavior rather than the internal traits [117]. Servant leaders make lonely followers feel understood through practical interactions. In contrast to perspective-taking, empathy’s behavioral aspects have rarely been discussed in the workplace. Lonely followers have a negative attitude towards socializing, and they avoid social interactions and conceal their loneliness, which results in a vicious cycle of loneliness [118]. As the risk of workplace loneliness rises, particularly post-epidemic, developing empathic communication skills has become a top priority for building effective leadership [119].

### 5.2. The Mediation Role of Colleagues’ Empathic Communication

This study clarified the mediating role of colleagues’ empathic communication between servant leaders and followers’ workplace loneliness, and servant leaders encourage all followers to engage in empathic communication to alleviate the suffering of lonely followers. Previous study indicates that servant leaders put the interests of their subordinates above their own [17]. Therefore, when colleagues of lonely followers receive helping behaviors from servant leaders, they might reciprocate the servant leaders by adopting organizational citizenship behaviors that benefit the leaders and their colleagues [86,96]. In addition, workplace loneliness is negatively associated with many organizational outcomes. In attempting to reduce the negative impacts that workplace loneliness has on the organization, colleagues of lonely followers will engage in more interpersonal helping behaviors with lonely followers [120]. To summarize, lonely followers receive more empathic communication from their colleagues, which prevents their workplace loneliness.

### 5.3. The Serial Mediation Roles of Leaders’ Empathic Communication and Colleagues’ Empathic Communication

We demonstrated that leaders’ empathic communication and colleagues’ empathic communication play a serial mediation role between servant leadership and followers’ workplace loneliness. This result corroborates the work of Hunter et al. (2013) [49], stating that servant leadership creates servant followers. Two explanations exist for the serial mediation effects of leaders’ empathic communication and colleagues’ empathic communication on workplace loneliness. First, servant leaders initiate the service cycle with empathic communication. They establish a service and inclusive climate in the organization [43,96] and develop followers’ prosocial values [121]. As a result, followers are willing and motivated to learn empathic communication skills from the servant leader to build relationships with lonely colleagues and alleviate their symptoms of loneliness. Second, social learning theory [89] emphasizes the importance of role modeling. Followers may recognize the leader as a role model [122]. After witnessing servant leaders’ empathic behavior toward lonely colleagues, followers will attempt to imitate servant leaders by acting on empathic behavior. In short, servant leaders reduce workplace loneliness not only by their behavior but also by guiding all their followers to adopt empathic communication actively.

### 5.4. Implications

This study provides theoretical support for reducing followers’ workplace loneliness. First, we determined the effectiveness of servant leadership in alleviating loneliness at work. Servant leadership is sensitive to the negative feelings of their followers [50]. Once lonely followers hide their lonely experiences and resort to negative coping strategies such as social avoidance, servant leaders can promptly identify these signals and use empathic behaviors to help followers get through the situation more quickly than other leadership styles.

Furthermore, we moved the focus on empathy from traits to behaviors. As an essential attribute of servant leadership, empathy embodies the core characteristic of service that distinguishes servant leadership from other leadership styles. Empathic communication emphasizes the perceptions of the target (lonely followers) rather than the observer (leader and colleagues), and perceptions of behavior may be more predictive of followers’ feelings of loneliness. However, current research on leader empathy almost focuses on leaders’ internal traits, with little attention paid to leaders’ empathic behaviors. Our study filled this research gap, and the results confirmed that empathic communication, as a tool for servant leaders [123], helps promote followers’ mental health.

Finally, the serial mediation roles of leaders’ empathic communication and colleagues’ empathic communication reveal that servant leadership can create an atmosphere of mutual help within the organization, where lonely followers can receive emotional support from both servant leaders and colleagues. This approach significantly reduces the stress leaders experience when managing interpersonal relationships at work. Organizations may contemplate a broader organizational culture [80] of empathy to improve employee well-being. Specifically, organizations can schedule regular one-on-one interviews between leaders and followers once a week to help leaders use empathic communication to prevent and promptly detect feelings of loneliness in followers. Leaders can also set aside free talk time in daily meetings to promote an empathic organizational climate.

### 5.5. Limitations and Future Directions

Readers should be conscious of some limitations to this study’s findings.

In terms of data collection, although the current data included employees from most provinces of China, the distribution of participants’ occupational categories was limited. The effectiveness of servant leadership in reducing followers’ workplace loneliness may vary by occupation and workplace. Therefore, future research could investigate more employees in different occupations and compare the effects of servant leadership on loneliness among employees in different occupations. In addition, we used a cross-sectional study design to confirm the hypothesized model. It is recommended that future research conduct a longitudinal analysis to examine the effects of servant leadership on followers’ workplace loneliness.

Second, this study measured servant leaders and leaders’ and colleagues’ empathic communication via self-report. While self-report data eliminate the likelihood of leaders and colleagues overestimating their level of empathic communication, they may also have resulted in recall bias. Future research could utilize a combination of self-report and other-report to assess empathic communication to ensure that the results are closest to the actual level of empathy.

Third, this study only considered the role of empathic communication in full-time office work. The COVID-19 epidemic altered the work model, and many organizations have adopted telework and hybrid work patterns. Further exploration of teleworkers’ workplace loneliness is needed. The COVID-19 pandemic changed work models, and many organizations have imported telework and hybrid work patterns. More research is needed into remote workers’ loneliness. The model of remote work may reduce the mediating effect of empathy. For example, the mitigating effects of using empathic communication in remote work (e.g., videoconferencing, instant messaging) may be weaker than in office work.

This study collected data in China. Considering cultural differences, in countries with a stronger tendency toward groupism (e.g., Japan), an interdependent construal of self may enhance the positive effects of empathic communication on workplace loneliness. On the contrary, the positive effect of empathic communication may become weaker in countries with stronger individualism. Future research could collect data from different countries to test whether the inhibitory effect of servant leadership on workplace loneliness is generally effective.

Finally, this study collected data in China. Considering cultural differences, in countries with stronger groupism (e.g., Japan), an interdependent construal of self may enhance the positive effects of empathic communication on workplace loneliness. On the contrary, the positive effect of empathic communication may weaken in countries with more robust individualism. Future research could collect data from different countries to test whether the inhibitory effect of servant leadership on workplace loneliness is generally effective.

## Figures and Tables

**Figure 1 behavsci-14-00004-f001:**
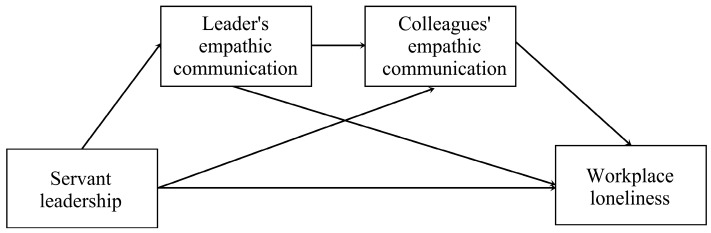
Hypothesized model.

**Figure 2 behavsci-14-00004-f002:**
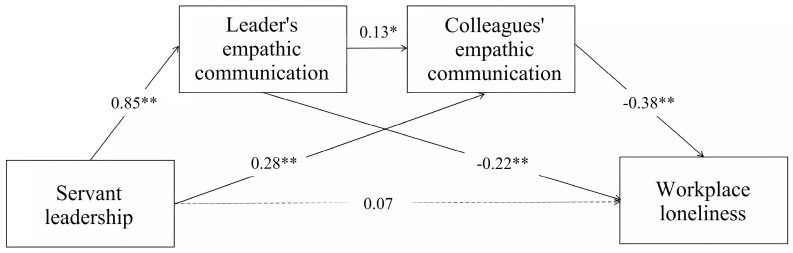
Serial mediation model shows effects on followers’ workplace loneliness. Note: * *p* < 0.05, ** *p* < 0.001.

**Table 1 behavsci-14-00004-t001:** Demographic characteristics of the participants.

Demographic Characteristics		N/M	Frequency (%)
Age		31.5	
Gender	Male	159	59.6
	Female	108	40.4
Education	College and below	58	21.8
	Undergraduate	190	71.2
	Master or above	19	7.1
Tenure (years)	<5	86	32.1
	5–10	142	53.2
	>5	39	14.6
Company size	<50	23	8.6
	50–100	79	29.6
	101–500	101	37.8
	>500	64	24
Position	General employee	91	34.1
	Junior manager	120	44.9
	Middle manager	54	20.2
	Senior manager	2	0.7
working hours (per week)	<40	49	18.4
	40–50	204	76.4
	>50	14	5.2
Total		267	100.0

**Table 2 behavsci-14-00004-t002:** Results of confirmatory factor analysis.

	Mean	SD	1	2	3	4
1. Servant leadership	3.82	0.75	-			
2. Leader’s empathic communication	3.98	0.80	0.80 **	-		
3. Colleagues’ empathic communication	4.32	0.50	0.60 **	0.56 **	-	
4. Workplace loneliness	1.62	0.44	−0.50 **	−0.59 **	−0.64 **	-

Note: ** *p* < 0.01. SD: standard deviation.

**Table 3 behavsci-14-00004-t003:** Results of hierarchical regression analysis.

Variables	Leader’s Empathic Communication		Colleagues’ Empathic Communication		Workplace Loneliness	
	Model 1	Model 2	Model 3	Model 4	Model 5	Model 6
Constant	3.69 **	0.87 **	4.44 **	3.05 **	1.80 **	4.03 **
Age	−0.01	−0.01	−0.02 *	−0.02 **	0.01	0.01
Tenure (years)	0.03	0.01	0.05 **	0.04 **	−0.04 **	−0.02 **
Position	0.19 **	0.02	0.06	−0.02	−0.12 **	−0.07 *
Servant leadership		0.85 **		0.28 **		0.07
Leader’s empathic communication				0.13 *		−0.22 **
Colleagues’ empathic communication						−0.38 **
R^2^	0.05	0.64	0.11	0.44	0.16	0.53
F	4.92	114.70	11.31	40.54	16.28	48.03

Note: * *p* < 0.05. ** *p* < 0.01.

**Table 4 behavsci-14-00004-t004:** Bootstrap 95% confidence intervals for mediation pathways.

Path	Effect	BootSE	Boot LLCL	Boot ULCL	Ratio of Indirect Effects
Total indirect effect	−0.35	0.06	−0.470	−0.249	1
Path 1: Servant leadership → Leader’s empathic communication → Workplace loneliness	−0.19	0.06	−0.304	−0.082	52.40%
Path 2: Servant leadership → Colleagues’ empathic communication → Workplace loneliness	−0.12	0.03	−0.183	−0.061	33.59%
Path 3: Servant leadership → Leader’s empathic communication → Colleagues’ empathic communication → Workplace loneliness	−0.05	0.03	−0.115	−0.003	14.01%
Path 1–Path 2	−0.07	0.07	−0.334	0.110	
Path 1–Path 3	−0.14	0.06	−0.418	−0.025	
Path 2–Path 3	−0.07	0.05	−0.300	0.040	

Note: BootSE: bootstrap standard error. BootLLCI: bootstrap lower-limit confidence interval. BootUULCI: bootstrap upper-limit confidence.

## Data Availability

The data presented in this study are available upon request from the corresponding author.

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
