# Peer review of "The Role of Empathic Communication in the Relationship between Servant Leadership and Workplace Loneliness: A Serial Mediation Model"

_behavsci, 2023, doi:10.3390/bs14010004_

Round 1

Reviewer 1 Report

Comments and Suggestions for Authors

Dear Authors,

you can find my comments attached!

All the best!

Comments on the Quality of English Language

English is fine.

Reviewer 2 Report

Comments and Suggestions for Authors

Thank you for submitting your manuscript to Behavioral Sciences. I have now completed my review and, regrettably, I must recommend that your manuscript be rejected for publication in its current form. .

  1. 1. Study addresses an interesting and relevant topic. However, the conceptual framework lacks a comprehensive integration of key theories and concepts. For instance, the linkage between servant leadership and workplace loneliness is not adequately grounded in existing literature.

2. The role of empathic communication as a mediator is an intriguing aspect but requires further clarification and support from empirical studies.

3. The methodology section lacks detailed information on sample selection, data collection procedures, and the rationale behind the choice of statistical methods used for the serial mediation model.

4. There is a need for a more rigorous justification of the chosen model, especially in the context of the hypotheses being tested.

5. The results section is somewhat confusing and does not clearly present the findings in a manner that supports the proposed serial mediation model.

6. The discussion section requires a more critical evaluation of the results, including potential limitations of the study and implications for future research.

7. The practical implications of the findings for organizations and leaders are mentioned but not explored in depth.

Comments on the Quality of English Language

The manuscript would benefit from a thorough proofreading to correct grammatical errors and improve sentence structure for better readability.

Reviewer 3 Report

Comments and Suggestions for Authors

Dear authors,

Thank you for submitting your manuscript to our journal. After careful consideration, I believe that the paper titled "The Role of Empathic Communication in the Relationship between Servant Leadership and Workplace Loneliness: A Serial Mediation Model" holds valuable insights into the role of leadership styles in addressing workplace loneliness. However, I recommend minor revisions to enhance the clarity and depth of the paper before it can be considered for publication. My comments and suggestions are as follows:

  1. 1. Clarification of Conceptual Framework: While your study interestingly integrates social learning theory and social exchange theory into the analysis of servant leadership, there could be greater clarity in how these theories specifically underpin the mechanisms you propose. A more detailed explanation of how these theories inform the relationship between servant leadership, empathic communication, and workplace loneliness would strengthen the theoretical foundation of your research.

  2.  
  3. 2. Expanding Literature Review: Your literature review effectively sets the stage for your research but could benefit from a broader scope. Consider incorporating recent studies that explore similar themes in diverse cultural or organizational contexts. This would not only enrich the context of your study but also help in highlighting the novelty of your research.

  4.  
  5. 3. Methodology Details: While you have outlined your methodology, providing more details about the survey design and the measures used for servant leadership and empathic communication could enhance the replicability of your study. Additionally, elaborating on the sampling strategy and addressing any potential biases in sample selection would enhance the robustness of your methodology section.

  6.  
  7. 4. Discussion of Limitations: Your manuscript would benefit from a more thorough discussion of its limitations. While all studies have constraints, acknowledging and discussing these limitations transparently can provide readers with a balanced view of the research. For instance, the cultural specificity of your sample (employees from 28 provinces in China) might limit the generalizability of your findings.

  8.  
  9. 5. Implications and Future Research Directions: Your conclusion effectively summarizes the key findings and implications of your research. However, it could be enriched by suggesting specific future research directions. For instance, exploring how servant leadership and empathic communication function in different types of organizational cultures or in virtual work environments could be valuable extensions of your work.

  10.  
  11. 6. Formatting and References: Please ensure that the manuscript adheres to the journal's formatting guidelines, especially in the references section. Consistency in formatting enhances the professional presentation of your research.

  12.  
  13. 7. Clarity and Language: While the paper is generally well-written, there are a few instances where clearer language could be used for better understanding. I suggest a careful proofreading to correct minor grammatical errors and improve sentence structure for enhanced clarity and readability.

In summary, your study provides meaningful contributions to the understanding of how servant leadership can alleviate workplace loneliness through empathic communication. With the suggested minor revisions, I believe your manuscript will be a valuable addition to our journal.

Round 2

Reviewer 2 Report

Comments and Suggestions for Authors

The revised manuscript can be accepted for publication.